# Skillful Kilometer-Scale Regional Weather Forecasting via Global and Regional Coupling

## Abstract

Data-driven weather models have advanced global medium-range forecasting, yet high-resolution regional prediction remains challenging due to unresolved multiscale interactions between large-scale dynamics and small-scale processes such as terrain-induced circulations and coastal effects. This paper presents a **global-regional coupling framework** for kilometer-scale regional weather forecasting that synergistically couples a pretrained Transformer-based global model with a high-resolution regional network via a novel bidirectional coupling module, **ScaleMixer**. ScaleMixer dynamically identifies meteorologically critical regions through adaptive key-position sampling and enables cross-scale feature interaction through dedicated attention mechanisms. The framework produces forecasts at $0.05°$ ($\sim 5$km) and 1-hour resolution over China, significantly outperforming operational NWP and AI baselines. It exhibits exceptional skill in capturing fine-grained phenomena such as orographic wind patterns, Foehn warming, and coastal transitions during typhoon events, demonstrating effective global-scale coherence with high-resolution fidelity. The code is available at `https://anonymous.4open.science/r/ScaleMixer-6B66`.

## 1 Introduction

Accurate weather forecasting is essential for disaster mitigation, agriculture, transportation, and energy management (Coiffier, 2011). Traditional numerical weather prediction (NWP) systems solve the governing equations of atmospheric dynamics involving mass continuity, momentum conservation, and thermodynamics, and parameterize subgrid-scale processes such as turbulence and cloud microphysics (Hurrell et al., 2013; Bouallègue et al., 2024). Although NWP models provide physically consistent forecasts and remain operational standards, their computational demands and sensitivity to parameterization schemes limit the skill in resolving kilometer-scale weather phenomena governed by multiscale interactions.

Recent data-driven AI models, particularly Transformer-based architectures trained on global re-analysis data such as ERA5, have achieved remarkable success in medium-range forecasting at synoptic scales at resolution of $0.25°$ and coarser. However, high-resolution operational regional forecasting (e.g., $0.05°$, or $\sim 5$ km) remains a significant challenge. Kilometer-scale weather is governed by complex multiscale interactions: large-scale circulations modulate local processes such as topographic flows, coastal breezes, and convective systems, while fine-scale features also feedback to broader dynamics. A prime example is the Hengduan Mountains, where large-scale dynamics including the Indian Monsoon, East Asian Monsoon, and Tibetan Plateau climate, interact with extreme terrain gradients. These terrain gradients, which exceed $3,000$ m within $100$ km, drive localized wind accelerations, sharp temperature contrasts, and convective processes that are poorly captured by coarse global models or isolated regional models (Xiang et al., 2024). Such intricate multiscale interactions challenge conventional models, necessitating forecasting models that reconcile global-scale coherence with high-resolution fidelity.

Recent studies have begun to explore data-driven regional weather forecasting and downscaling, typically treating global forecasts as static inputs (Nipen et al., 2024; Oskarsson et al., 2023; Xu et al.; Qin et al., 2024; Mardani et al., 2025). However, these decoupled methods neglect dynamic cross-scale interactions and suffer from temporal misalignment between low-frequency global forecasts (e.g., 6-hourly) and high-resolution regional observations (e.g., hourly). In summary, to make accurate

high-resolution regional weather forecasting requires addressing two key challenges: (1) *a mechanism to dynamically identify regions where cross-scale interactions are active*, and (2) *a bidirectional coupling framework that ensures spatial-temporal consistency across scales*.

To address the aforementioned challenges, we propose a novel **global–regional coupling framework** for high-resolution regional weather prediction. Our approach seamlessly integrates a pretrained global Transformer model, which provides synoptic-scale (large scale) context, with a regional refinement model operating at $0.05°$ resolution. Central to this architecture is **ScaleMixer**, a module that adaptively identifies key spatial regions exhibiting strong multiscale interactions and enable bidirectional feature encoding between global and regional tokens. This allows the model to prioritize meteorologically critical areas such as typhoon boundaries and mountain ridges, and maintain global coherence while resolving fine-grained regional dynamics. The main contributions of this work are summarized as follows:

- A **global–regional coupling framework for** $0.05°$ **and** $1$**-hour forecasting** by integrating a pretrained global model for synoptic-scale context with a high-resolution regional model
- The **ScaleMixer** module for dynamic identification of cross-scale interaction regions and bidirectional feature fusion.
- Extensive evaluation over complex terrain and coastal zones in China, demonstrating state-of-the-art performance against operational NWP and leading AI baselines, with notable skill in capturing orographic wind effects, Foehn warming, and typhoon boundary-layer transitions.

## 2 RELATED WORKS

**Numerical Weather Forecasting**  As the predominant paradigm, NWP systems typically formulate the atmospheric physical laws through PDEs and then solve them using numerical simulations. Representative examples include earth system models (ESMs) (Hurrell et al., 2013) and the operational Integrated Forecast System (IFS) of European Centre for Medium-Range Weather Forecasts (ECMWF) (Bouallègue et al., 2024). By integrating physics laws, NWP approaches have enjoyed remarkable success with great accuracy, stability, and interpretability. IFS-HRES (European Centre for Medium-Range Weather Forecasts, 2023) is a world-leading high-resolution deterministic NWP system ($0.1°$) and serves as a benchmark for operational forecasting and research. However, NWP models are sensitive to initial conditions, prone to errors in parameterization, and computationally expensive (Kochkov et al., 2024). These limitations hinder their ability to accurately resolve kilometer-scale weather driven by complex multiscale interactions.

**Deep Learning for Global Weather Forecasting**  Recent progress in deep learning models for global weather forecasting has been transformative. They predominantly employ two architectural paradigms: Transformer-based models (Niu et al., 2025; Chen et al., 2023a; Bi et al., 2023; Chen et al., 2023b) and Graph Neural Network (GNN)-based architectures (Keisler, 2022; Lam et al., 2023b; Price et al., 2023). These models demonstrate computational efficiency and competitive skill in predicting synoptic-scale weather patterns. However, they fail to capture finer-grained mesoscale weather dynamics due to limited resolution ($0.25°$ or coarser).

**Deep Learning for Regional Weather Forecasting and Downscaling**  Recently, regional weather models have been developed for fine-scale forecasting and downscaling over regions of interest. For instance, CorrDiff (Mardani et al., 2025) combines U-Net and diffusion models to correct and downscale the global forecasts to improve local predictions. Limited area modeling methods (Nipen et al., 2024; Gao et al., 2025) employ GNN architectures with stretched-grid and nested-grid to make global weather forecasts. They model cross-scale interactions via grid deformation and nesting, and such rigid, geometric interactions limits the model's ability to capture highly dynamic and non-local coupling processes efficiently. On the other hand, our model employ a content-adaptive cross-scale interation mechanism.

## 3 METHODOLOGY

Accurate regional weather forecasting requires seamless integration of large-scale atmospheric dynamics with localized, high-resolution features As nearly all AI-based global models are trained on the EAR5 dataset, we assume a pretrained Vision Transformer (ViT)-based global weather

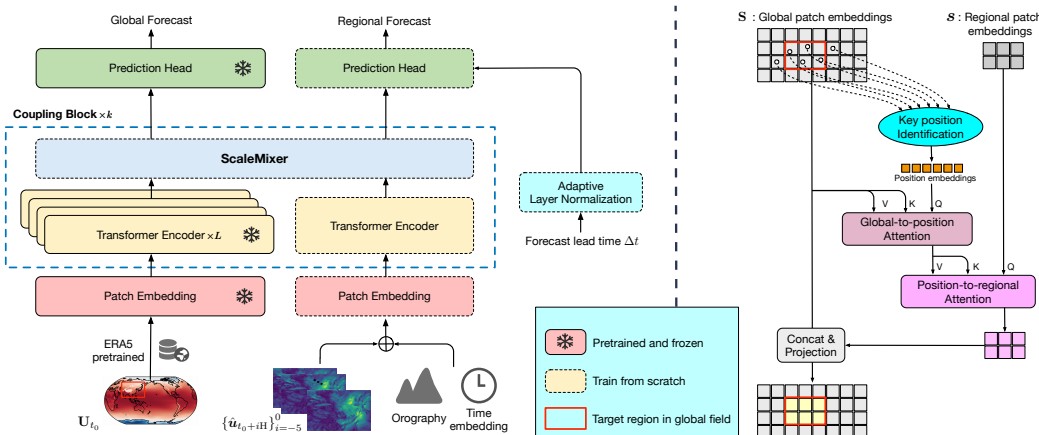

Figure 1: **Left:** The Architecture of Global-Regional Weather Forecasting Model: Synoptic-scale context ($\mathcal{M}_{\text{global}}$) drives mesoscale regional refinement ($\mathcal{M}_{\text{regional}}$) via ScaleMixer, ensuring cross-scale coupling and consistency. **Right:** ScaleMixer Module: Bidirectional Cross-Scale Coupling via Key Position Identification and encoding. Key components include (1) key position identification, (2) coupling regional dynamics with global context via global-to-position and position-to-regional attention, and (3) global token adaptation incorporating regional features.

forecasting model, denoted $\mathcal{M}_{\text{global}}$, which operates on low-resolution ($0.25°$) global reanalysis data $\mathbf{U}^{t_0} \in \mathbb{R}^{H \times W \times C}$. At time $t_0$, the model generates 6-hour-ahead global predictions capturing synoptic-scale dynamics:

$$\hat{\mathbf{U}}^{t_0+6\text{H}} = \mathcal{M}_{\text{global}}(\mathbf{U}^{t_0}),\tag{1}$$

where $H \times W \times C$ represents the stacked weather state with multiple levels of upper air and surface variables, in which latitude and longitude are divided into $H$ and $W$ grids for each variable. Concurrently, high-resolution regional analysis data $\boldsymbol{u}^{t_0} \in \mathbb{R}^{h \times w \times V_{\text{reg}}}$ provides critical surface variables (wind components $U, V$, temperature $T$, specific humidity $Q$, pressure $P$, radiation fluxes $SSRD$, and total cloud cover $TCC$ ) within a region of interest at $1\,\text{hour}$ temporal resolution and $0.05°$ spatial resolution.

**Problem Formulation**  As the fundamental challenge lies in effectively coupling multiscale information: coarse-grained global features from $\mathcal{M}_{\text{global}}$ and fine-resolution regional features, we formalize the task as developing a hybrid global-regional weather forecasting framework $\mathcal{M}_{\text{global}-\text{regional}}$ that extends $\mathcal{M}_{\text{global}}$ through the integration of large-scale atmospheric dynamics and small-scale weather effects.

$$\hat{\mathbf{U}}^{t_0+6\text{H}}; \left\{\hat{\boldsymbol{u}}^{t_0+i\text{H}}\right\}_{i=1}^{6} = \mathcal{M}_{\text{global}-\text{regional}}\left(\mathbf{U}^{t_0}; \left\{\boldsymbol{u}^{t_0+i\text{H}}\right\}_{i=-5}^{0}\right),\tag{2}$$

where $\left\{\hat{\boldsymbol{u}}^{t_0+i\text{H}}\right\}_{i=-5}^{0}$ denotes the temporally aligned regional analysis data with 1-hour intervals. This formulation establishes a principled framework for generating high-fidelity regional forecasts by systematically bridging global-scale dynamics with localized meteorological processes with deep learning architectures.

**Model Overview**  We propose a multiscale weather forecasting framework that dynamically integrates global and regional-scale atmospheric dynamics to resolve high-resolution mesoscale features in the target region. As shown in Figure 1, the framework comprises two Transformer-based submodels with shared architectural principles: (1) a global model for synoptic-scale dynamics, and (2) a regional model for mesoscale processes. The ScaleMixer module enables bidirectional coupling between the global and regional models via adaptive key position identification and encoding to preserve cross-scale meteorological consistency. The global model, pretrained on ERA5 reanalysis data (Hersbach et al., 2020), remains fixed during regional optimization, while the regional model and ScaleMixer module are trained from scratch.

### 3.1 PRETRAINED GLOBAL WEATHER MODEL

Our global model $\mathcal{M}_{\text{global}}$ prioritizes architectural simplicity, flexibility, and scalability, and implements a vision Transformer (ViT) architecture (Dosovitskiy et al., 2020). Without loss of generality, our framework can work with any ViT-based global weather forecasting model. In the following discussion, we will limit the discussion on our in-house developed ViT-based global model, comprising three core components:

**Patch Embedding and Tokenization:** A 2-D convolutional layer partitions the multivariate input atmospheric state $\mathbf{U}^{t_0} \in \mathbb{R}^{H \times W \times C}$ into non-overlapping spatial patches of size $P \times P$. This generates token representations $\mathbf{S} \in \mathbb{R}^{N \times d}$, where $N = (H/P) \times (W/P)$ and $d$ is the embedding dimension.

**Transformer Encoder:** A stack of $M$ Transformer encoder layers processes the sequence $\mathbf{S}$ through multi-head self-attention and feed-forward networks (Vaswani et al., 2017; Dosovitskiy et al., 2020), enabling global information interaction across spatial scales.

**Prediction Head:** A deconvolution block upscales the processed sequence back to the original spatial resolution $H \times W$, producing a 6-hour ahead deterministic global forecast $\mathbf{U}^{t_0+6\text{H}}$ of the full atmospheric state.

The global model $\mathcal{M}_{\text{global}}$ is pretrained on ERA5 reanalysis (Hersbach et al., 2020) using weighed mean absolute error (MAE) as the loss function (detailed in Section 3.4). The dataset includes five pressure level variables (13 vertical levels each): geopotential ($z$), specific humidity ($q$), wind components ($u, v$), and temperature (t), and multiple surface variables, e.g., 2-meter temperature (t2m), 10-meter wind (u10, v10), and mean sea level pressure (msl), surface pressure (sp), etc. (detailed in Appendix B).

### 3.2 MODIFICATIONS IN REGIONAL WEATHER MODEL

The regional model $\mathcal{M}_{\text{regional}}$ inherits the Transformer architecture from the global model but introduces necessary modifications: (1) modified patch embedding layer to incorporate fine-grained topography and temporal encodings, (2) enhanced prediction head with adaptive layer normalization (AdaLN) (Peebles & Xie, 2023) to amplify the high-frequency signal for hourly temporal alignment, and (3) fewer Transformer encoder layers ($k \ll M$) to reduce computational overhead while preserving regional meteorological fidelity.

**Patch Embedding:** In addition to the input $\left\{ \boldsymbol{u}^{t_0+i\text{H}} \right\}_{i=-5}^{0} \in \mathbb{R}^{h \times w \times V_{\text{reg}} \times 6}$, the block also needs to process the static topography, land-sea mask, and dynamic hourly temporal information. Regional analyses are tokenized across 6 time steps using a shared patch embedding layer, with topography, land-sea masks, and temporal embeddings (hour-of-day, day-of-year) added via MLP. To ensure geographic consistency with global patches, we set patch size $p = 5 \times P$, generating regional tokens $\mathbf{s} \in \mathbb{R}^{n \times d}$, where $n = (h/p) \times (w/p)$.

**Transformer Encoders:** The regional model employs $k$ encoder layers ($k \ll M$, where $M = k \times L$) to achieve computational efficiency in regional optimization. Each cross-scale coupling block comprises $L$ global encoder layers, 1 regional encoder layer, and 1 ScaleMixer module.

**Prediction Head:** To generate 6-hour forecasts at hourly intervals, 6 dedicated prediction heads produce lead time-specific outputs ($\Delta t = 1\text{H}$ to $6\text{H}$). Temporal alignment is enforced via AdaLN(Peebles & Xie, 2023), where scale and shift parameters $\gamma, \beta$ are derived from Fourier embeddings of $\Delta t$:

$$\text{FourierEmbed}(\Delta t) = [\cos(2\pi a_i \Delta t + b_i), \sin(2\pi a_i \Delta t + b_i)] \text{ for } 0 \leq i < d/2, \qquad (3)$$
$$\gamma, \beta = \text{MLP}\left(\text{FourierEmbed}(\Delta t)\right), \qquad (4)$$

where $a_i$ and $b_i$ are learnable Fourier embedding parameters. This formulation ensures high-frequency signal amplification for regional forecasting. Moreover, regional prediction heads take the concatenation of regional tokens and spatially-aligned global tokens as input to make full use of multi-scale information.

## 3.3 ScaleMixer: Bidirectional Global and Regional Scale Coupling

Accurate high-resolution regional prediction requires resolving multiscale atmospheric processes–from synoptic-scale forcings to mesoscale circulations–while maintaining global dynamical consistency. To this end, we introduce **ScaleMixer**, a differentiable coupling mechanism that explicitly models interactions between the global foundation model and the regional refinement model. As illustrated in Figure 1 (right), ScaleMixer enables bidirectional feature fusion by adaptively identifying meteorologically critical regions and performing token-level encoding, effectively prioritizing areas with strong cross-scale interactions.

**Adaptive key position identification** To capture spatial regions exhibiting strong multiscale interactions, we implement a dynamics-aware sampling module that identifies critical spatial positions from global token embeddings $\mathbf{S}$. Spatial dynamics are extracted via a convolutional network, followed by softmax-normalized importance scores $\mathbf{Pr} \in \mathbb{R}^N$ ($N$ is the number of global tokens):

$$\mathbf{Pr} = \text{Softmax}\left(\text{Conv}(\mathbf{S})\right), \tag{5}$$

where $\text{Conv}(\cdot)$ consists of a convolutional layer followed by a linear projection. We then select top-$m$ salient positions:

$$\mathbf{c} = \arg\text{top-}m(\mathbf{Pr}), \quad \mathbf{h} = \mathbf{Pr}[\mathbf{c}] \odot \mathbf{S}[\mathbf{c}], \tag{6}$$

with $\odot$ denoting element-wise product, $\mathbf{c} = \{\mathbf{c}_i\}_{i=1}^m \in \mathbb{R}^{m \times 2}$ ($\mathbf{p}_i \in [0 : H/P - 1] \times [0 : W/P - 1]$) representing the coordinates of $m$ selected tokens, and $\mathbf{h} \in \mathbb{R}^{m \times d}$ their corresponding embeddings.

**Regional features alignment with global context** To effectively bridge the scale gap between global context and regional features, we design a two-stage cross-attention mechanism operating on identified key positions. Directly correlating all global and regional tokens is computationally expensive and may weaken localized meteorological features. Instead, we first condense global information into a sparse set of dynamically identified key positions, then propagate these enriched features to regional tokens.

*Global-to-Position Attention* first aggregates global context into the key positions. Using the concatenated token embeddings and coordinates of key positions $\mathbf{h}||\mathbf{c} \in \mathbb{R}^{m \times (d+2)}$ as queries, and the global tokens $\mathbf{S}$ as keys and values, we compute:

$$\text{Glo-to-Pos}(\mathbf{h}||\mathbf{c}, \mathbf{S}, \mathbf{S}) = \text{Softmax}\left(\frac{(\mathbf{W}_Q \cdot \mathbf{h}||\mathbf{c})(\mathbf{W}_K \mathbf{S})^\top}{\sqrt{d}}\right)\mathbf{W}_V \mathbf{S}, \tag{7}$$

$$\mathbf{h}_{\text{global}}||\mathbf{c}' = \mathbf{h}||\mathbf{c} + \text{Glo-to-Pos}(\mathbf{h}||\mathbf{c}, \mathbf{S}, \mathbf{S}), \tag{8}$$

where $\mathbf{W}_Q$, $\mathbf{W}_K$ and $\mathbf{W}_V$ are linear projections. To better model the dynamics of key positions, the key representations are further refined by incorporating regional features via bilinear interpolation at the updated coordinates $\mathbf{c}'$

$$\mathbf{h}' = \text{MLP}_{\text{Proj}}\left(\text{Bilinear}(\mathbf{s}, \mathbf{c}')||\mathbf{h}_{\text{global}}\right). \tag{9}$$

*Position-to-regional attention* subsequently integrates the globally informed key features into regional tokens $\mathbf{s}$:

$$\text{Pos-to-Reg}(\mathbf{s}, \mathbf{h}'||\mathbf{c}', \mathbf{h}'||\mathbf{c}') = \text{Softmax}\left(\frac{\mathbf{W}'_Q \mathbf{s}(\mathbf{W}'_K \cdot \mathbf{h}'||\mathbf{c}')^\top}{\sqrt{d}}\right)\mathbf{W}'_V \cdot \mathbf{h}'||\mathbf{c}', \tag{10}$$

$$\mathbf{s}' = \mathbf{s} + \text{Pos-to-Reg}(\mathbf{s}, \mathbf{c}'||\mathbf{p}', \mathbf{c}'||\mathbf{p}'). \tag{11}$$

with distinct learnable projections $\mathbf{W}'_Q$, $\mathbf{W}'_K$, and $\mathbf{W}'_V$. This two-step attention mechanism ensures that synoptic-scale dynamics are effectively integrated into high-resolution regional features, enabling globally consistent and locally accurate weather prediction.

**Global token adaptation with regional feedback** To enable large-scale dynamics to adapt to regional details, global tokens spatially aligned with regional tokens ($\mathbf{S}_{\text{aligned}}$) are updated via token-wise concatenation and an adapter MLP:

$$\mathbf{S}'_{\text{aligned}} = \text{Concat}\left(\mathbf{S}_{\text{aligned}}, \mathbf{s}'\right) \in \mathbb{R}^{n \times 2d}, \tag{12}$$

$$\mathbf{S}''_{\text{aligned}} = \mathbf{S}_{\text{aligned}} + \text{MLP}_{\text{Adapter}}\left(\mathbf{S}'_{\text{aligned}}\right). \tag{13}$$

These adapted tokens $\mathbf{S}''_{\text{aligned}}$ replace their counterparts in the global token sequence, allowing regional fine-scale information to recursively influence the global context in subsequent encoder layers.

## 3.4 Model Optimization

The optimization schedule follows a three-stage training protocol: (1) global model pretraining, (2) regional model one-step training (6-hours ahead), and (3) regional model autoregressive roll-out fine-tuning ($12 \sim 48$-hours ahead).

**Training objective**   For global model pretraining, we employ the weighted mean absolute error (MAE) across multivariate atmospheric states. Decomposing the weather state $\mathbf{U}^t$ into surface-level variables and upper-air atmospheric variables, $\hat{\mathbf{U}}^t = (\hat{\mathbf{S}}^t, \hat{\mathbf{A}}^t)$ and $\mathbf{U}^t = (\mathbf{S}^t, \mathbf{A}^t)$, the loss can be written as:

$$
\mathcal{L}(\hat{\mathbf{U}}, \mathbf{U}^t) = \frac{1}{V_S + V_A} \left[ \left( \sum_{k=1}^{V_S} \frac{w_k^S}{H \times W} \sum_{i=1}^{H} \sum_{j=1}^{W} |\hat{\mathbf{S}}_{i,j,k}^t - \mathbf{S}_{i,j,k}^t| \right) \\
+ \left( \sum_{k=1}^{V_A} \frac{1}{H \times W \times P} \sum_{p=1}^{P} w_{c,k}^A \sum_{i=1}^{H} \sum_{j=1}^{W} |\hat{\mathbf{A}}_{i,j,p,k}^t - \mathbf{A}_{i,j,p,k}^t| \right) \right],
\tag{14}
$$

where $V_A$ and $V_K$ are numbers of upper-air and surface varibles, $P$ is the number of pressure levels, $w_k^S$ is the weight associated with surface-level variable $k$, and $w_{k,c}^A$ is the weight associated with atmospheric variable $k$ at pressure level $p$.

During both one-step training and roll-out fine-tuning of regional model, we directly using MAE as the objective:

$$
\mathcal{L}(\hat{\boldsymbol{u}}, \boldsymbol{u}^t) = \frac{1}{V_{\text{reg}}} \left( \sum_{k=1}^{V_{\text{reg}}} \frac{1}{h \times w} \sum_{i=1}^{h} \sum_{j=1}^{w} |\hat{\boldsymbol{u}}_{i,j,k}^t - \boldsymbol{u}_{i,j,k}^t| \right),
\tag{15}
$$

where $V_{\text{reg}}$ is the number of variables in regional analyses.

**Implementation and Training Details**   The global Transformer encoder comprises 24 layers ($M = 24$), while the regional encoder and ScaleMixer modules each contain 4 layers ($k = 4$). The model employs a hidden dimension of 384 and identifies $m = 64$ key positions for cross-scale interaction in each ScaleMixer module. The framework contains 1.07 billion parameters, with the global model $\mathcal{M}_{\text{global}}$ accounting for 736 million. Full implementation details are summarized in Appendix A.

The global model was pretrained for $150,000$ steps on $32\times$ NVIDIA A800 GPUs using the AdamW optimizer (Loshchilov & Hutter, 2017) with a per-GPU batch size of 1. A cosine learning rate schedule was applied with linear warmup over 1,000 steps, decaying from $7 \times 10^{-4}$ to $1 \times 10^{-7}$. Regional model training followed identical hyperparameters over $80,000$ iterations on $8\times$ A800 GPUs, with $\mathcal{M}_{\text{global}}$ parameters frozen. During regional roll-out fine-tuning, the model was trained for $100,000$ steps at a fixed learning rate of $1 \times 10^{-6}$.

## 4 Experiments

To resolve high-impact meteorological phenomena such as convective storms and boundary layer dynamics, weather prediction systems require high-resolution spatial-temporal modeling capabilities. We evaluate ScaleMixer through two complementary experimental paradigms: (1) ***hindcast*** for verification using reanalysis data, and (2) ***operational forecast*** to assess predictive skill under dynamically evolving initial conditions consistent with production environment management system.

### 4.1 Datasets

**Global Reanalysis (ERA5)**   The European Centre for Medium-Range Weather Forecasts (ECMWF) ERA5 reanalysis provides $0.25°$ horizontal resolution ($1440 \times 720$ latitude-longitude grid) atmospheric states with 37 hybrid pressure levels. Spanning 1979–2015, this dataset serves as the

Table 1: Average RMSE and ACC across $\Delta t = 1 \sim 48$ hours lead time regional weather hindcast at $0.05°$ resolution. The best results are **bolded**.

| Variable | Latitude-weighted RMSE | | | | | | | | Latitude-weighted ACC | | | | | | |
|---|---|---|---|---|---|---|---|---|---|---|---|---|---|---|---|
| | IFS-HRES | Baguan | $\mathcal{M}_{\text{global}}$ | $\mathcal{M}_{\text{regional}}$ | OneForecast | LAM | ScaleMixer | $\Delta$RMSE | IFS-HRES | $\mathcal{M}_{\text{global}}$ | $\mathcal{M}_{\text{regional}}$ | OneForecast | LAM | ScaleMixer | $\Delta$ACC |
| T2M | 1.815 | 1.928 | 1.991 | 2.452 | 1.571 | 1.742 | **1.382** | ↓23.86% | 0.862 | 0.881 | 0.845 | 0.897 | 0.901 | **0.921** | ↑6.84% |
| U10 | 1.934 | 1.956 | 1.967 | 2.631 | 2.251 | 1.889 | **1.644** | ↓14.99% | 0.721 | 0.753 | 0.716 | 0.744 | 0.742 | **0.793** | ↑9.99% |
| V10 | 1.928 | 1.937 | 1.970 | 2.886 | 2.316 | 1.905 | **1.617** | ↓16.13% | 0.723 | 0.751 | 0.713 | 0.732 | 0.737 | **0.785** | ↑8.57% |
| Q | 0.807 | 0.611 | 0.811 | 1.243 | 0.714 | 0.676 | **0.559** | ↓30.73% | 0.774 | 0.768 | 0.771 | 0.751 | 0.781 | **0.812** | ↑4.91% |
| P | 13.27 | 22.97 | 10.27 | 4.241 | 2.545 | 2.142 | **1.874** | ↓85.88% | 0.887 | 0.902 | 0.894 | 0.899 | 0.912 | **0.920** | ↑3.72% |
| TCC | 38.83 | 31.83 | (N/A) | 43.71 | 37.74 | 34.13 | **28.76** | ↓25.93% | 0.563 | (N/A) | 0.617 | 0.621 | 0.679 | **0.721** | ↑28.06% |
| SSRD | 51.25 | 41.18 | (N/A) | 68.42 | 52.56 | 42.41 | **33.26** | ↓34.04% | 0.824 | (N/A) | 0.834 | 0.838 | 0.850 | **0.887** | ↑7.64% |

**(1)** $\Delta$RMSE and $\Delta$ACC donote RMSE and ACC improvement of ScaleMixer compared to IFS-HRES. **(2)** $\mathcal{M}_{\text{global}}$ denotes standalone global model, and $\mathcal{M}_{\text{regional}}$ denotes uncoupled regional model. **(3)** Results of IFS-HRES ($0.1°$) and Baguan, and $\mathcal{M}_{\text{global}}$ ($0.25°$) are corrected and downscaled to target grid ($0.05°$) using a **pretrained bias-correction and downscaling model** (based on a ViT backbone trained on ERA5 and CLDAS data) for comparison. **(4)** T2M: 2m temperature; U10/V10: 10m wind components; Q: Specific humidity; P: Surface Pressure; TCC: Total cloud cover; SSRD: Radiation flux (surface solar radiation downward).

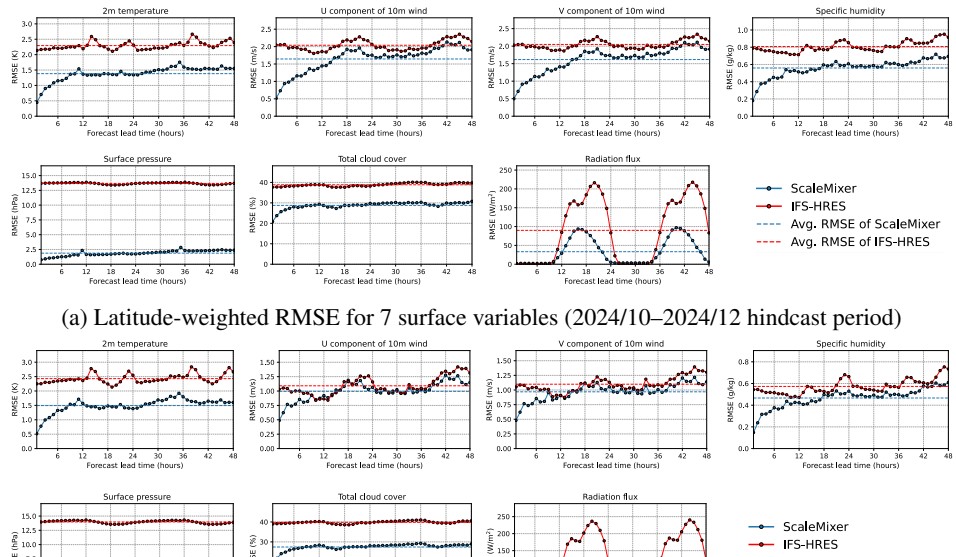

(a) Latitude-weighted RMSE for 7 surface variables (2024/10–2024/12 hindcast period)

(b) Latitude-weighted RMSE for 7 surface variables (2025/01–2025/04 operational period)

Figure 2: **ScaleMixer demonstrates superior deterministic forecasting skill compared to IFS-HRES at $0.05°$ resolution.** Seven surface variables (T2M, U10, V10, Q, P, TCC, and SSRD) are evaluated using latitude-weighted RMSE (lower values indicate superior performance). **(a)** Hindcast results show ScaleMixer outperforms IFS-HRES across all variables during 2024/10–2024/12. **(b)** Operational forecasts confirm ScaleMixer maintains superiority performance (2025/01–2025/04).

primary training source for the global model ($\mathcal{M}_{\text{global}}$), with 2016 reserved for validation. ERA5's spatiotemporal continuity and multivariate fidelity make it a standard for data-driven weather modeling (Hersbach et al., 2020).

**Global Operational Analysis** Operational analysis utilize the initial conditions from ECMWF's High-Resolution Deterministic Prediction (HRES) system, which assimilates observations through 4D-variational data assimilation (Rabier & Liu, 2003). The $0.1°$ analysis fields (interpolated to ERA5 resolution, $0.25°$) provide dynamically real-time initial conditions for ScaleMixer's operational deployment during 2025/01–2025/04.

**Regional Analysis (CLDAS)** The China Meteorological Administration's Land Data Assimilation System (CLDAS) offers $0.01°$ resolution meteorological fields over East Asia (0–65°N, 60–160°E) with surface variables critical for regional forecasting. We employ CLDAS data (interpolate to $0.05°$) from 2022/01–2024/09 for global-regional model ($\mathcal{M}_{\text{global-regional}}$) training, with two independent evaluation periods defined as: ***Hindcast evaluation*** (ERA5 input): 2024/10–2024/12 and ***Operational evaluation*** (operational analysis input): 2025/01–2025/04.

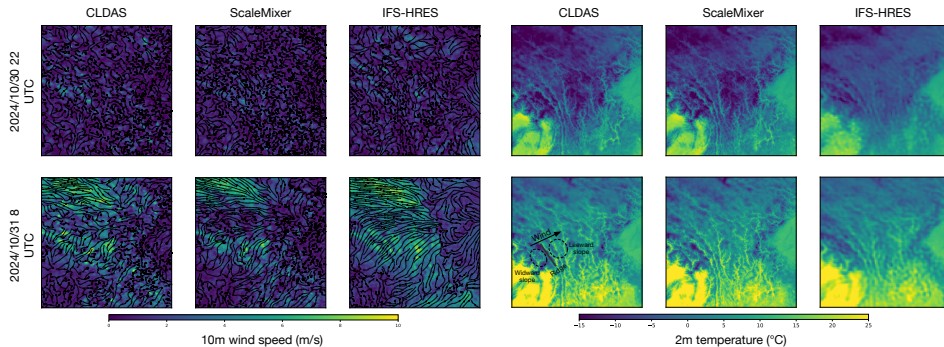

Figure 3: **Left:** Temporal evolution of 10m wind speed predictions initialized at 2024/10/30 12 UTC over the Hengduan Mountains (25.0–35.0°N, 95.0–105.0°E), China. Black arrows represent wind flow fields. ScaleMixer resolves enhanced resolution of orographic wind heterogeneity (peaking >10 m/s at crests and <2 m/s in valleys). **Right:** Corresponding temperature fields. Foehn effects are illustrated in the picture, characterized by 4–8°C leeward warming relative to windward slopes through adiabatic compression processes. ScaleMixer captures fine-grained temperature gradients, contrasting with IFS-HRES exhibiting spatial smoothing forecasts.

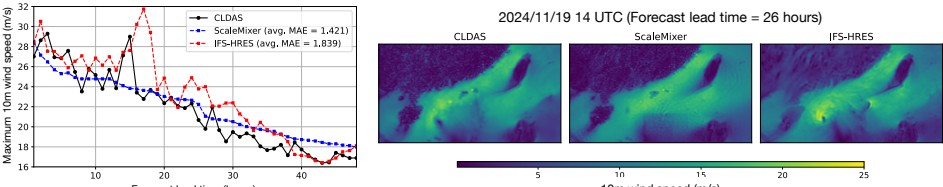

Figure 4: **Left:** Maximum 10m wind speed predictions during Typhoon Man-yi landfall (from 2024-11-18 13 UTC to 2024-11-20 12 UTC) from ScaleMixer (0.05°) and IFS-HRES (0.1°). ScaleMixer demonstrates superior capability in resolving abrupt wind speed reductions near landfall. **Right:** Coastal wind field predictions (16.3–26.3°N, 106.5–125.0°E) for Typhoon Man-yi showing ScaleMixer's (0.05°) ability to capture high-resolution sea-land wind transitions compared to CLDAS ground truth.

More details of datasets and experimental settings can be found in Appendix B.

## 4.2 EVALUATION METRICS AND BASELINES

**Evaluation metrics**    To measure the performance of regional weather forecasting, we evaluate all methods using latitude-weighted root mean squared error (RMSE) and latitude-weighted anomaly correlation coefficient (ACC). More details of metrics can be found in Appendix C.

**Baselines**    We comprehensively evaluate ScaleMixer against several strong baselines: (1) our internal global model ($\mathcal{M}_{\text{global}}$); (2) a standalone regional model initialized from CLDAS data without global coupling ($\mathcal{M}_{\text{regional}}$); (3) an AI-based global forecast model Baguan (Niu et al., 2025), which demonstrates superior performance among a set of state-of-the-art data-driven weather models on WeatherBench [1] and provides more comprehensive surface meteorological variables than other global models like Pangu-Weather (Bi et al., 2023) and GraphCast (Lam et al., 2023a); (4) the operational high-resolution NWP system IFS-HRES from ECMWF (ECMWF, 2023), which serves as a gold-standard reference. The resolutions of AI-based global forecasts and IFS-HRES are 0.25° and 0.1°, respectively, and there may exist systematic bias between their forecast values and CLDAS. For a fair comparison, we employ a downscaling and correction model to map the original forecast values to the target 0.05° grids. The downscaling and correction model is trained on ERA5 and CLDAS, using Swin Transformer (Liu et al., 2021) as the backbone, containing 4 layers block and a hidden dimension of 192; (5) OneForecast Gao et al. (2025), which introduces a Neural Nested Grid method that typically passes boundary feature maps between grids of different resolutions via direct

---
[1]https://sites.research.google/gr/weatherbench/scorecards-2020/

interpolation and concatenation; and (6) Limited Area Model (LAM), which is built upon $\mathcal{M}_{regional}$ (Standalone Regional Model) but enhanced to take both regional initial conditions and external global forecasts as input.

### 4.3 Skillful Regional Weather Forecasting at 0.05° Resolution

We focus on short-term forecasting for next 48 hours, primarily because such outlooks have a more immediate impact on societal functions and daily routines. Furthermore, this is the period that NWP models have optimal performances. The deterministic forecasting results of ScaleMixer and baselines are summarized in Table 1, and Figure 2, evaluating forecast skill across $\Delta t = 1 \sim 48$ hours lead time.

**Hindcast evaluation**  For the ERA5-driven hindcast period (2024/10–2024/12), ScaleMixer achieves significant improvements across all seven surface variables (T2M, U10, V10, Q, P, TCC, and SSRD) compared to both standalone global/regional baselines and IFS-HRES (Table 1), verifying the effectiveness of coupling global and regional scales. ScaleMixer achieves 40.86% lower latitude-weighted RMSE and 9.96% higher ACC compared to IFS-HRES, indicating enhanced resolution capability for mesoscale convective systems and boundary layer dynamics. As shown in Figure 2a, performance advantages persist consistently across forecast horizons.

**Operational forecast evaluation**  Under dynamically evolving operational initial conditions (2025/01–2025/04), ScaleMixer maintains superior skill despite real-time analysis field uncertainties (Figure 2b). Compared to IFS-HRES at 0.1° resolution, statistically significant RMSE improvements are sustained through 48-hour lead times under operational constraints, with pronounced improvements in 1–24-hours ahead predictions where regional-scale processes dominate.

### 4.4 Case Studies

**Orographic-induced wind and temperature**  As exemplified in Figure 3 (left) for wind prediction of the complex terrain regions in the Hengduan Mountains (25.0–35.0°N, 95.0–105.0°E) China, ScaleMixer (0.05°) resolves wind characteristics across topographic gradients: maximum wind speed at mountain crests (exceeding 10 m/s) and deceleration within valleys (<2 m/s). This contrasts with IFS-HRES (0.1°) which exhibits systematic underestimation of orographic wind characteristics due to insufficient subgrid-scale orographic parametrization.

Moreover, the same orographic forcing that generates wind heterogeneity also drives temperature variations. ScaleMixer resolves pronounced temperature contrasts across elevation gradients (Fig. 3, right), with leeward slopes exhibiting 4–8°C warming relative to windward sides, a canonical Foehn effect signature[2], arising from adiabatic compression of descending air masses. In contrast, IFS-HRES underestimates these temperature gradients, failing to capture dependencies between terrain steepness and temperature variation. The enhanced resolution with data-driven method in ScaleMixer enables superior representation fine-grained weather features in complex terrain.

**Extreme event prediction**  We conduct a extreme event prediction case study of Typhoon Man-yi, a high-impact tropical cyclone which took place across East Asia in late 2024. We initialize forecasts on 11/18 12 UTC and validate against IFS-HRES. ScaleMixer (0.05°) accurately predict the observed 10m wind speed reduction during landfall transitions. Compared to IFS-HRES (0.1°), the enhanced resolution preserves sharper sea-land breeze contrasts and mesoscale convective structures critical for extreme wind predictions, demonstrating the advantage of data-driven downscaling in resolving cyclone dynamics.

Additional visualizations of forecasts are provided in Appendix F, which demonstrate the framework's capability to capture high-resolution meteorological details.

### 4.5 Ablation Studies

To rigorously validate the architectural design of ScaleMixer, we conducted fine-grained ablation studies focusing on two core dimensions: the sampling strategy for key position identification and the directionality of cross-scale coupling. Furthermore, we analyzed the sensitivity of the model

---

[2]https://en.wikipedia.org/wiki/Foehn_wind

to critical hyperparameters. All ablation experiments were conducted on the validation set with a forecast lead time of $\Delta t = 24$ hours.

**Effectiveness of ScaleMixer Components.** We compared our proposed framework against four variants: (A) *Random Sampling*, replacing adaptive identification with random selection; (B) *Fixed Uniform Grid*, utilizing a static grid for interaction; (C) *Unidirectional Coupling*, allowing only global-to-regional information flow; and (D) *No Interaction*, equivalent to the standalone regional model. The results are summarized in Table 2.

Table 2: Ablation study of ScaleMixer components on 48-hour forecast performance.

| Model Variant | Configuration Details | T2M RMSE | U10 RMSE |
|---|---|---|---|
| **ScaleMixer** | **Adaptive Sampling + Bidirectional** | **1.382** | **1.644** |
| Variant A | Random Sampling + Bidirectional | 1.605 (+16.1%) | 1.882 (+14.5%) |
| Variant B | Fixed Uniform Grid + Bidirectional | 1.512 (+9.4%) | 1.795 (+9.2%) |
| Variant C | Adaptive Sampling + Unidirectional | 1.468 (+6.2%) | 1.721 (+4.7%) |
| Variant D | No Interaction (Standalone) | 1.991 (+44.1%) | 1.967 (+19.6%) |

The results demonstrate: (1) **Effectiveness of Adaptive Sampling:** The proposed adaptive key position identification significantly outperforms the Fixed Uniform Grid (Variant B), reducing T2M RMSE by 9.4%. This demonstrates that dynamically focusing computation on meteorologically active regions (e.g., high-gradient boundaries) is far more efficient than uniform processing, which may waste capacity on static areas; and (2) **Effectiveness of Bidirectional Coupling:** Compared to unidirectional coupling (Variant C), our bidirectional mechanism achieves a 6.2% improvement in T2M RMSE. This confirms that allowing high-resolution regional features to explicitly refine global tokens creates a necessary closed-loop feedback, enhancing the consistency of the synoptic-scale context.

**Hyperparameter Sensitivity.** We further investigated the sensitivity of the regional encoder depth ($k$). As shown in Table 4, increasing the number of layers from $k = 2$ to $k = 4$ yields significant gains, while $k = 8$ offers diminishing returns with doubled computational cost. Based on these results, we adopted $k = 4$ as the default configuration to balance forecasting accuracy and inference efficiency.

Table 3: Sensitivity analysis of Regional Encoder Layers ($k$).

| Metric | Regional Encoder Layers ($k$) | | |
|---|---|---|---|
| | $k = 2$ | $k = 4$ | $k = 8$ |
| T2M RMSE | 1.485 | 1.382 | 1.379 |
| U10 RMSE | 1.752 | 1.644 | 1.641 |
| *Inference Time* | *22ms* | *28ms* | *51ms* |

## 5   CONCLUSION

In this paper, we present a multiscale deep learning framework for high-resolution regional weather forecasting that bridges synoptic-scale dynamics with localized mesoscale processes. By integrating a pretrained global foundation model and a novel bidirectional global-regional coupling module, ScaleMixer achieves state-of-the-art performance in resolving complex weather phenomena at $0.05°$ ($\sim 5\,\text{km}$) resolution. Experimental results establish ScaleMixer as a robust data-driven approach for regional weather forecasting, particularly with complex terrain and coastal dynamics. In the future, we will extend the framework to probabilistic forecasting and assimilate multi-modal observations (e.g., radar, satellite) for real-time forecasting.

## 6   ETHICS STATEMENT

As our work only focuses on the weather forecasting problem, there is no potential ethical risk.

# 7 REPRODUCIBILITY STATEMENT

In the main text, we have formally defined the model architecture with equations. All the implementation details, including dataset descriptions, metrics, and experiment configurations, are provided in the manuscript and the code (available online).

# 8 DECLARATION OF LLM USAGE

The author of this paper only used LLM as a grammar checker and simple text polishing tool. LLM was not used in any of the ideas or technical implementations.

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

## A IMPLEMENTATION DETAILS

In our multiscale regional weather forecasting framework, the backbones of the global model ($\mathcal{M}_{\text{global}}$) and regional model ($\mathcal{M}_{\text{regional}}$) are based on ViT (Dosovitskiy et al., 2021). The framework contains 1.07 billion parameters, with the global model $\mathcal{M}_{\text{global}}$ accounting for 736 million. The hyperparameter configurations of the model are summarized in Table 4.

Table 4: Default hyperparameters of the framework.

| Module | Hyperparameter | Description | Value |
|---|---|---|---|
| Global Model $\mathcal{M}_{\text{global}}$ | $P$ | Patch size of global tokens | 6 |
| | $d$ | hidden dimension | 384 |
| | $M$ | Number of Transformer encoder layers of in the global model | 24 |
| | Heads | Number of attention heads | 8 |
| | MLP ratio | Expansion factor for MLP | 4. |
| | Depth of prediction head | Number of deconvolution layers of the final prediction head | 2 |
| | Drop path | Stochastic depth rate | 0.1 |
| | Dropout | Dropout rate | 0.1 |
| Regional Model $\mathcal{M}_{\text{regional}}$ | $p$ | Patch size of regional tokens | 30 |
| | $d$ | hidden dimension | 384 |
| | $k$ | Number of Transformer encoder layers of in the regional model | 4 |
| | Heads | Number of attention heads | 8 |
| | MLP ratio | Expansion factor for MLP | 4. |
| | Depth of prediction head | Number of deconvolution layers of the final prediction head | 2 |
| | Drop path | Stochastic depth rate | 0.1 |
| | Dropout | Dropout rate | 0.1 |
| ScaleMixer | Depth | total number of ScaleMix modules | 4 |
| | Depth of position identification block | number of convolution layers in position identification block | 1 |
| | Kernel size | kernel size of convolution layers in position identification block | 3 |
| | $m$ | number of key positions | 64 |

## B DATASET DETAILS AND EXPERIMENTAL SETTINGS

In our experiments, we use the preprocessed **ERA5** data from WeatherBench (Rasp et al., 2020). EAR5 is a well-acknowledged weather forecasting benchmark dataset and it is widely used in data-driven weather forecasting methods. WeatherBench processed the raw ERA5 dataset[3], which includes 8 atmospheric variables across 13 pressure levels, 6 surface variables, and 5 static variables. We normalize all the inputs via z-score normalization for each variable at each pressure level. Also, we apply the inverse normalization for the predictions of future states for performance evaluation.

We collected and processed operational analysis data, which are used for operational forecast, from initial conditions of ECMWF's High-Resolution Deterministic Prediction (HRES) system, assimilating observations with 4D-variational data assimilation. The 0.1° analysis fields (interpolated to ERA5 resolution, 0.25°) provide dynamically real-time initial conditions. We set and process the atmosphere variables consistent with the ERA5 Dataset.

We also collected and processed the China Meteorological Administration's Land Data System data (CLDAS), which offers 0.01° resolution meteorological fields over East Asia (0–65°N, 60–160°E). The regional analysis dataset is used to train and evaluate regional weather forecasting model. The dataset includes 7 critical surface variables: wind components (U, V), temperature (T), specific humidity (Q), pressure (P), radiation fluxes (SSRD), and total cloud cover (TCC). We normalize all the inputs via z-score normalization for each variable at each pressure level. Also, we apply the inverse normalization for the predictions of future states for performance evaluation.

---

[3]More details of ERA5 data can be found in `https://confluence.ecmwf.int/display/CKB/ERA5%3A+data+documentation`.

## B.1 ERA5 AND OPERATIONAL ANLYSIS WITH 0.25° RESOLUTION

we selected 6 atmospheric variables at all **13 pressure levels**, 3 surface variables, and 3 static variables for the ERA5 dataset with 0.25° resolution, as detailed in Table 5. In our model training, we choose all variables as input variables, and all variables except three static variables as output variables that are used for loss calculation to pretrain global model $\mathcal{M}_{\text{global}}$.

Table 5: Summary of ECMWF variables utilized in the ERA5 and operational analysis dataset with 0.25° resolution. The variables $lsm$ and $oro$ are constant and invariant with time.

| Type | Variable Name | Abbrev. | Description | Pressure Levels |
|---|---|---|---|---|
| Static Variable | Land-sea mask | $lsm$ | Binary mask distinguishing land (1) from sea (0) | N/A |
| | Orography | $oro$ | Height of Earth's surface | N/A |
| | Latitude | $lat$ | Latitude of each grid point | N/A |
| Surface Variable | 2 metre temperature | $t2m$ | Temperature measured 2 meters above the surface | Single level |
| | 10 metre U wind component | $u10$ | East-west wind speed at 10 meters above the surface | Single level |
| | 10 metre V wind component | $v10$ | North-south wind speed at 10 meters above the surface | Single level |
| | Mean sea level presure | $msl$ | Pressure of the atmosphere adjusted to the height of mean sea level | Single level |
| | Surface pressure | $sp$ | Pressure of the atmosphere on the surface of land, sea and in-land water | Single level |
| | 2 metre dewpoint temperature | $d2m$ | Temperature to which the air, at 2 metres above the surface of the Earth | Single level |
| Upper-air Variable | Geopotential | $z$ | Height relative to a pressure level | 50, 100,150, 200, 250,300, 400, 500, 600, 700, 850, 925,1000 hPa |
| | U wind component | $u$ | Wind speed in the east-west direction | 50, 100,150, 200, 250,300, 400, 500, 600, 700, 850, 925,1000 hPa |
| | V wind component | $v$ | Wind speed in the north-south direction | 50, 100,150, 200, 250,300, 400, 500, 600, 700, 850, 925,1000 hPa |
| | Temperature | $t$ | Atmospheric temperature | 50, 100,150, 200, 250,300, 400, 500, 600, 700, 850, 925,1000 hPa |
| | Specific humidity | $q$ | Mixing ratio of water vapor to total air mass | 50, 100,150, 200, 250,300, 400, 500, 600, 700, 850, 925,1000 hPa |
| | Relative humidity | $r$ | Humidity relative to saturation | 50, 100,150, 200, 250,300, 400, 500, 600, 700, 850, 925,1000 hPa |

## B.2 CLDAS WITH 0.05° RESOLUTION

We selected 7 surface variables for the CLDAS dataset with 0.25° resolution, as detailed in Table 6. In our model training, we choose all variables as input variables, and all variables as output variables that are used for loss calculation to train global-regional model $\mathcal{M}_{\text{global}-\text{regional}}$.

Table 6: Summary of variables utilized in CLDAS with 0.05° resolution.

| Type | Variable Name | Abbrev. | Description |
|---|---|---|---|
| Surface Variable | 2 metre temperature | $T$ | Temperature measured 2 meters above the surface |
| | 10 metre U wind component | $U$ | East-west wind speed at 10 meters above the surface |
| | 10 metre V wind component | $V$ | North-south wind speed at 10 meters above the surface |
| | Surface specific humidity | $Q$ | Mixing ratio of water vapor to total air mass at 2 meters above the surface |
| | Surface pressure | $P$ | Pressure of the atmosphere on the surface of land, sea and in-land water |
| | Total cloud cover | $TCC$ | Cloud occurring at different model levels through the atmosphere |
| | Radiation flux (surface solar radiation flux downwards) | $SSRD$ | Flux of solar radiation that reaches a horizontal plane at the surface of the Earth |

## C EVALUATION METRICS FOR REGIONAL WEATHER FORECASTING

This section provides detailed explanations of all the evaluation metrics for regional weather forecasting used in the main experiments. For each metric, $\boldsymbol{u}$ and $\hat{\boldsymbol{u}}$ represent the predicted and ground truth values, respectively, both shaped as $h \times w \times V_{\text{reg}}$, where $V_{\text{reg}}$ is the number of total weather factors, and $h \times w$ is the spatial resolution of latitude ($h$) and longitude ($w$). To account for the non-uniform grid cell areas, the latitude weighting term $\alpha(\cdot)$ is introduced.

**Latitude-weighted Root Mean Square Error (RMSE)** assesses model accuracy while considering the Earth's curvature. The latitude weighting adjusts for the varying grid cell areas at different latitudes, ensuring that errors are appropriately measured. Lower RMSE values indicate better model performance.

$$\text{RMSE} = \frac{1}{V_{\text{reg}}} \sum_{k=1}^{V_{\text{reg}}} \sqrt{\frac{1}{hw} \sum_{i=1}^{h} \sum_{j=1}^{w} \alpha(i) \left(\hat{\boldsymbol{u}}_{i,j,k} - \boldsymbol{u}_{i,j,k}\right)^2}, \ \alpha(i) = \frac{\cos(\text{lat}(i))}{\frac{1}{h} \sum_{i'=1}^{h} \cos\left(\text{lat}\left(i'\right)\right)}.$$

**Anomaly Correlation Coefficient (ACC)** measures a model's ability to predict deviations from the mean. Higher ACC values indicate better accuracy in capturing anomalies, which is crucial in meteorology and climate science.

$$\text{ACC} = \frac{\sum_{i,j,k} \hat{\boldsymbol{u}}'_{i,j,k} \boldsymbol{u}'_{i,j,k}}{\sqrt{\sum_{i,j,k} \alpha(h)(\hat{\boldsymbol{u}}'_{i,j,k})^2 \sum_{i,j,k} \alpha(h)(\boldsymbol{u}'_{i,j,k})^2}},$$

where $\boldsymbol{u}' = \boldsymbol{u} - C$ and $\hat{\boldsymbol{u}}' = \hat{\boldsymbol{u}} - C$, with climatology $C$ representing the temporal mean of the same period over the training set.

## D    LIMITATION AND FUTURE WORK

While ScaleMixer demonstrates significant advancements in regional weather forecasting, several limitations warrant further investigation:

**Physical Consistency Constraints:** The current framework relies on data-driven learning without explicit hard constraints from governing equations (e.g., Navier-Stokes or hydrostatic balance). This may lead to unphysical artifacts in long-term roll-outs, particularly under extreme dynamical regimes (e.g., supercell storms). Future work will integrate physics-informed regularization to enhance adherence to conservation laws.

**Data Assimilation Latency:** The current implementation uses fixed regional analysis cycles (1-hour intervals) but lacks real-time adaptive assimilation of high-frequency observations (e.g., radar, satellite radiances). Integrating online data assimilation with attention-based observation weighting will be critical for operational deployment.

**Uncertainty Quantification:** Deterministic forecasts from ScaleMixer do not quantify predictive uncertainty, limiting its utility for risk-sensitive applications. Probabilistic extensions through ensemble forecasting and deep generative models are planned to address this gap.

These improvements will bridge the remaining gaps between AI-driven hybrid models and operational numerical weather prediction systems, particularly for high-impact weather scenarios requiring both global coherence and kilometer-scale fidelity.

## E    BROADER IMPACTS

This research focuses on high-resolution regional weather forecasting, which has an essential influence on relevant fields such as energy, transportation, and agriculture. As an AI application for social good, our model boosts predictions for various weather factors such as temperature, wind speed, and radiation flux. It is essential to note that our work focuses solely on scientific issues, and we also ensure that ethical considerations are carefully taken into account. Thus, we believe that there is no ethical risk associated with our research.

## F    VISUALIZATION OF FORECASTS

To intuitively demonstrate the forecasting capacity of our model, we present the showcases of weather forecasting results in China and zoomed-in regions in Figures 5, 6, 7 8, 9, 10, 11, 12, 13, 14, 15, and 16.

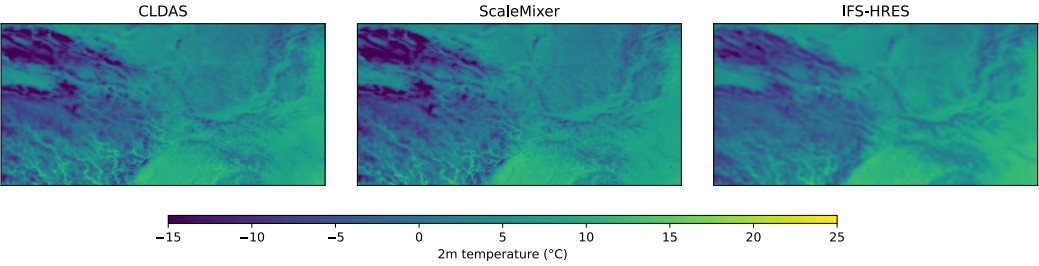

Figure 5: 2 metre temperature forecasts over China

Figure 6: 2 metre temperature forecasts over a subregion of latitudes in $[30, 40]$ and longitudes in $[95, 115]$

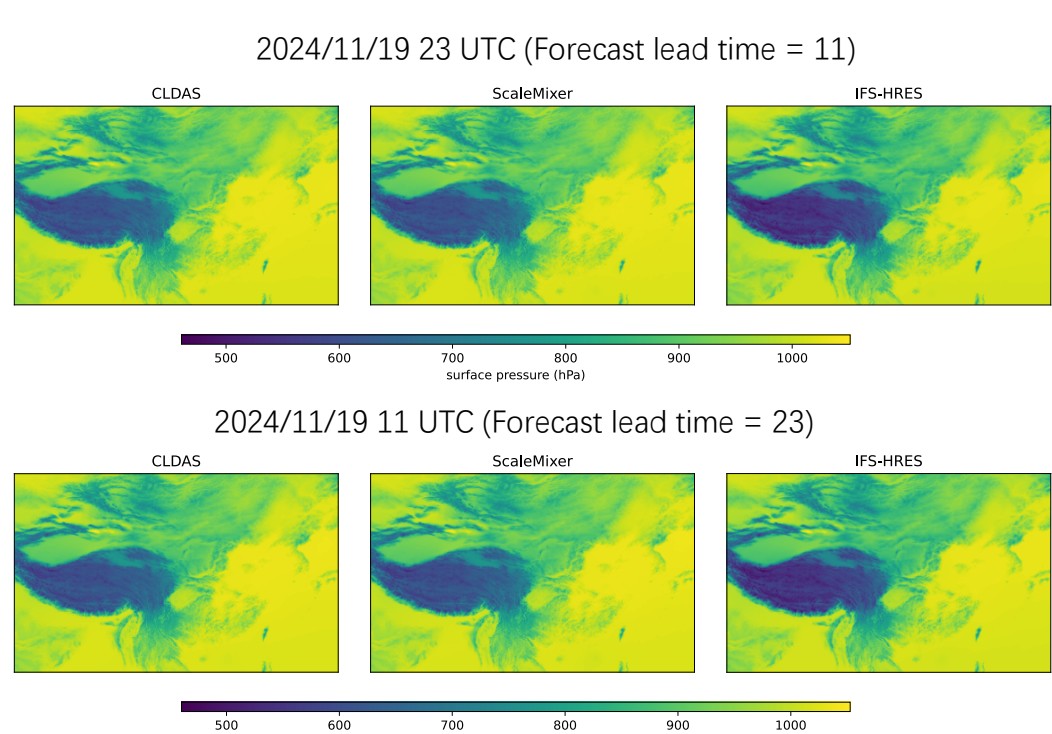

Figure 7: surface pressure forecasts over China

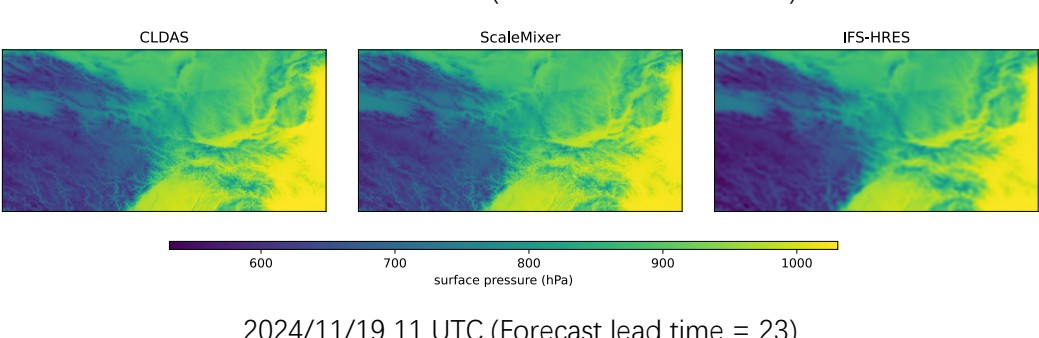

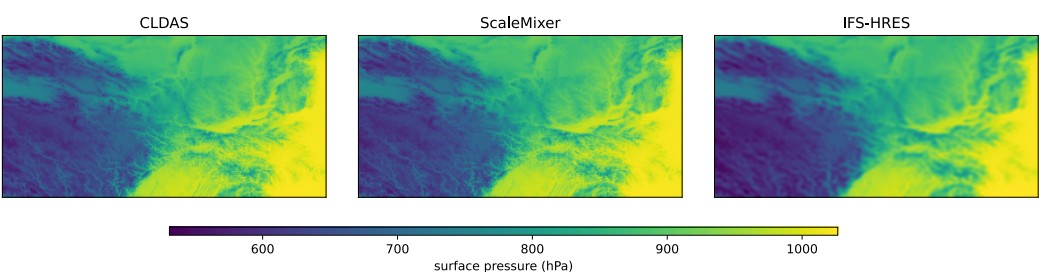

Figure 8: surface pressure forecasts over a subregion of latitudes in $[30, 40]$ and longitudes in $[95, 115]$

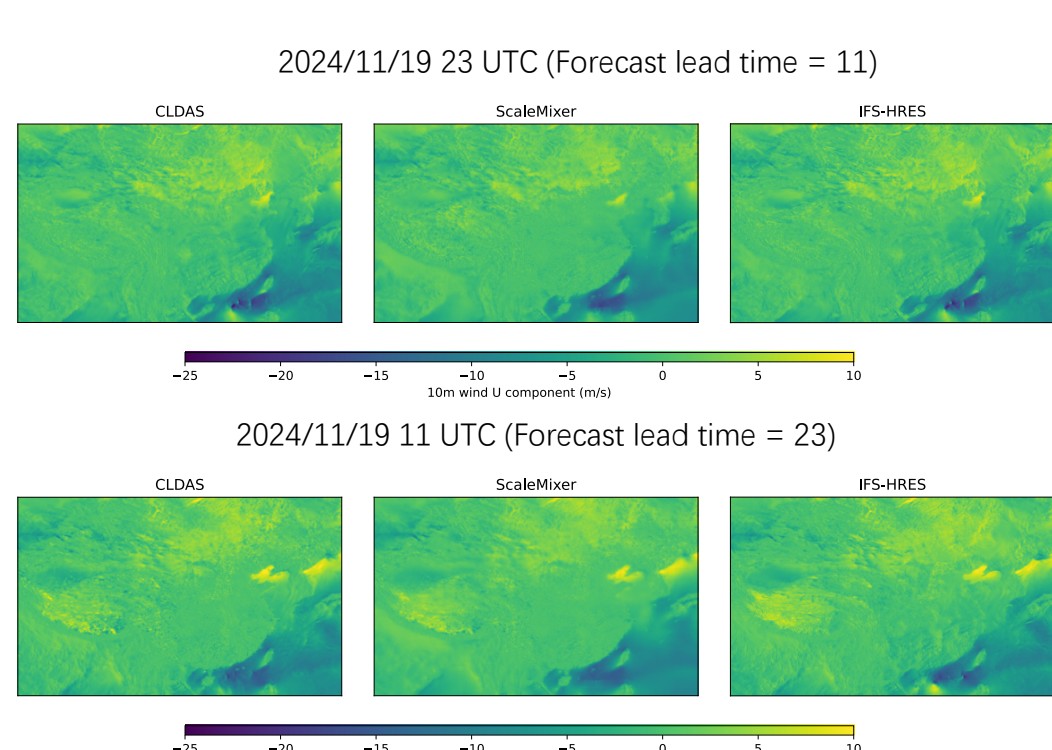

Figure 9: 10 metre Wind speed U component forecasts over China

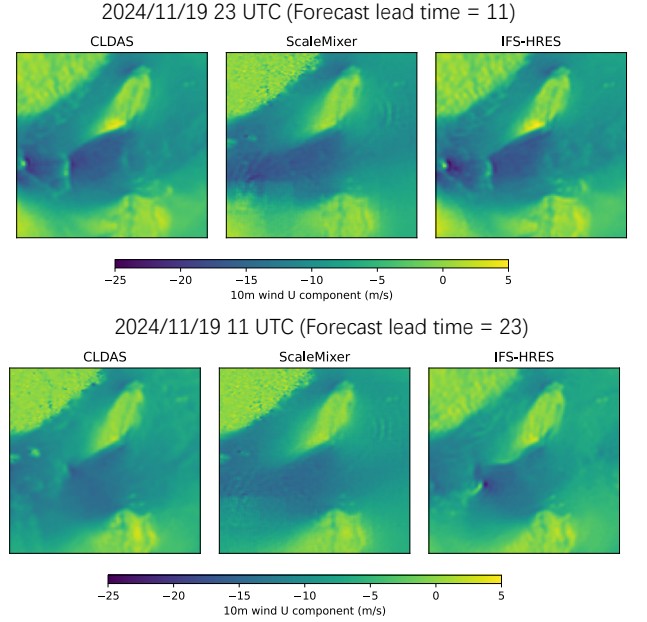

Figure 10: 10 metre wind speed U component forecasts over a subregion of latitudes in $[16.3, 26]$ and longitudes in $[115, 125]$

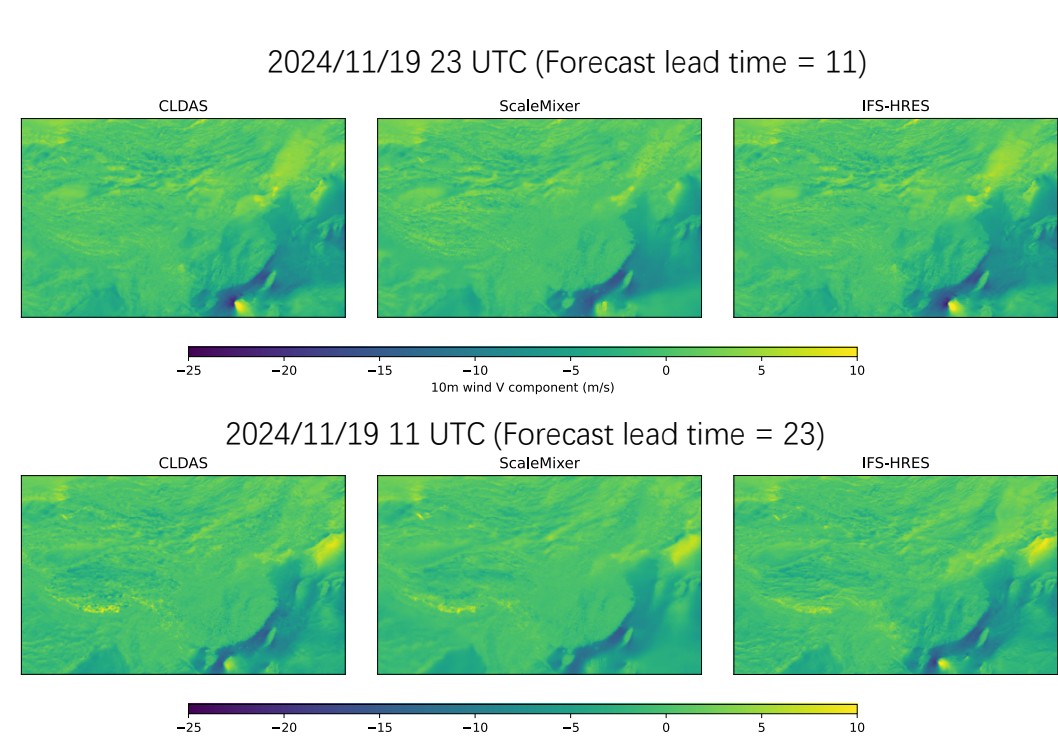

Figure 11: 10 metre Wind speed V component forecasts over China

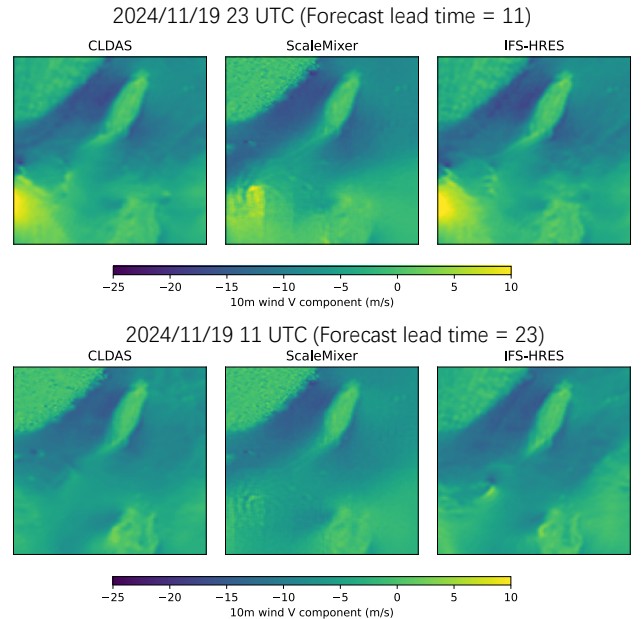

Figure 12: 10 metre wind speed V component forecasts over a subregion of latitudes in [16.3, 26] and longitudes in [115, 125]

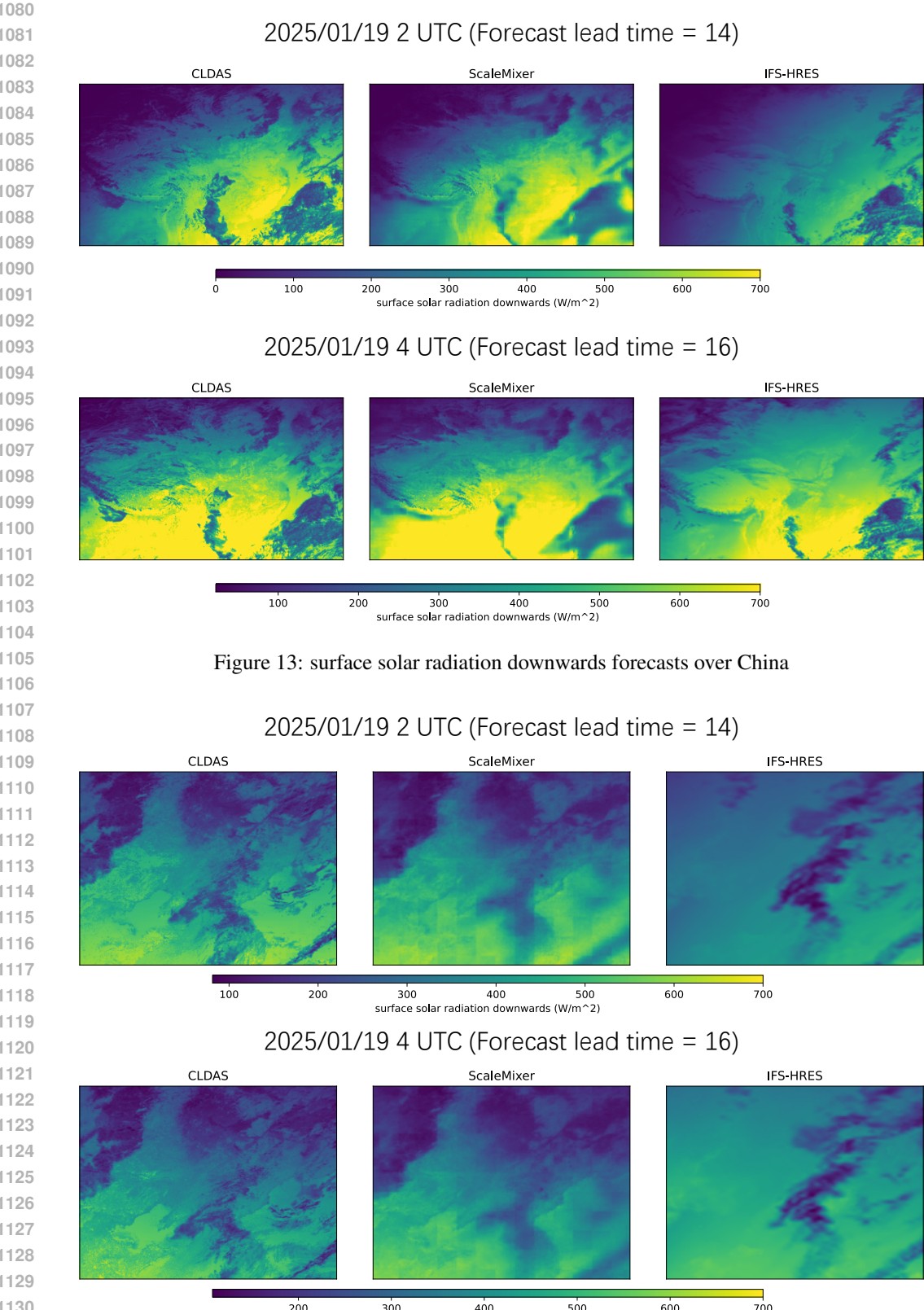

Figure 13: surface solar radiation downwards forecasts over China

Figure 14: surface solar radiation downwards forecasts over a subregion of latitudes in $[35, 50]$ and longitudes in $[115, 135]$

## 2025/01/18 13 UTC (Forecast lead time = 1)

## 2025/01/18 20 UTC (Forecast lead time = 8)

Figure 15: total cloud cover forecasts over China

## 2025/01/18 13 UTC (Forecast lead time = 1)

## 2025/01/18 20 UTC (Forecast lead time = 8)

Figure 16: total cloud cover forecasts over a subregion of latitudes in $[35, 50]$ and longitudes in $[115, 135]$

