# OpenReview forum: "Skillful Kilometer-Scale Regional Weather Forecasting via Global and Regional Coupling"
_ICLR.cc/2026/Conference — Submitted to ICLR 2026_

### Official Review · Reviewer_m4Re · 2025-10-22

**Soundness:** 2
**Presentation:** 3
**Contribution:** 3
**Rating:** 6
**Confidence:** 4

**Summary:**

The paper addresses the challenge of high-resolution regional weather forecasting, where global forecasts are typically used as inputs in a decoupled manner that overlooks cross-scale interactions, leading to less accurate predictions. To overcome this, the authors propose a global–regional coupling framework specifically designed for high-resolution regional forecasting. This framework includes a module called ScaleMixer, which identifies spatial regions with strong multiscale interactions and learns bidirectional feature encoding between global and regional tokens in these areas. The experimental analysis includes two complementary settings, hindcast and operational forecasting, both demonstrating the effectiveness of the proposed architecture.

Specifically, the authors formulate the forecasting problem as developing a hybrid global–regional weather forecasting framework that extends the capabilities of a pretrained global network by leveraging interactions between large-scale (global) atmospheric dynamics and small-scale (regional) weather effects. The pretrained global model, M_global, is a ViT-based architecture trained on the ERA5 reanalysis dataset. The regional model,  M_regional, uses the same architecture with certain modifications. The ScaleMixer module then models the interaction between these two networks by learning their relationships and performing token-level encoding based on them, effectively prioritizing regions with strong cross-scale dependencies.

**Strengths:**

- The authors clearly identify the gap in the current literature on regional weather forecasting. The related work section provides an up-to-date review of both global and regional forecasting studies. The motivation behind the proposed architecture is clearly presented as a mechanism to dynamically identify cross-scale interaction regions and enable bidirectional feature encoding between global and regional tokens through a coupling framework. The overall structure of the paper is clear, well-organized, and easy to follow.
- The experimental analysis (Table 1 and Figure 2) clearly demonstrates the superiority of the proposed model over other methods by a significant margin. Including the results of M_global and M_regional further highlights the need for a coupled architecture that leverages both global and regional information, hence the need for ScaleMixer. Case studies on orographic-induced wind and temperature forecasting, and extreme event prediction further support the effectiveness of the proposed approach. Visualizations of forecasts for various metrics, such as temperature and surface pressure, also show that ScaleMixer adapts well to different scenarios rather than performing well in only a single setting.

**Weaknesses:**

- For an experimental work like this, the ablation study is a crucial component, as it should strongly justify the need for each distinct part of the architecture. While the paper provides clear motivation for each component in earlier sections, for instance, the need for a ScaleMixer like structure, the choice of its exact architecture would be much more convincing if supported by a dedicated ablation study.
- Similarly, the proposed architecture includes multiple components that depend on various hyperparameters, such as patch size and the number of encoder layers. The authors neither specify what kind of validation—if any—was used to select these hyperparameters nor present results using alternative settings. Including these details would experimentally strengthen the evidence for the model’s efficiency and its superiority over other baseline methods.
- The authors state that the code for the proposed architecture and experiments is available at the provided URL for reproducibility. However, the URL does not appear to contain any implementation.

**Questions:**

- See the weakness section for my detailed comments.

---

> ### Author Response · Authors · 2025-11-21
> **Response to Reviewer m4Re**
>
> We sincerely thank you for constructive feedback. We appreciate your emphasis on the need for detailed ablation studies and hyperparameter justifications to ensure the rigor of our experimental results.
>
> Below, we address your concerns point-by-point, providing ablation comparisons and sensitivity analyses to further strengthen the soundness of our framework.
>
> ### 1. Ablation Study on ScaleMixer
>
> We agree that justifying the specific architecture designs is crucial. To demonstrate the necessity of the **Adaptive Key Position Identification** and **Bidirectional Coupling mechanisms** within ScaleMixer, we conducted fine-grained ablation studies focusing on two dimensions: **Sampling Strategy** (Adaptive vs. Random vs. Fixed Grid) and **Coupling Strategy** (Bidirectional vs. Unidirectional).
>
> We evaluated these variants on the 48-hour lead time forecasting. The results, summarized in the table below, strongly validate our design:
>
> **Table: Ablation Study of ScaleMixer Components (RMSE across $\Delta t = 1 \sim 48$ hours lead time)**
>
> | Model Variant | Sampling Strategy | Coupling Strategy | T2M | U10 |
> | :--- | :--- | :--- | :--- | :--- |
> | **ScaleMixer** | **Adaptive (Key Position)** | **Bidirectional** | **1.382** | **1.644** |
> |  A | Random Sampling | Bidirectional | 1.605 (+16.1%) | 1.882 (+14.5%) |
> |  B | Fixed Uniform Grid | Bidirectional | 1.512 (+9.4%) | 1.795 (+9.2%) |
> |  C | Adaptive (Key Position) | Unidirectional (Global $\to$ Regional) | 1.468 (+6.2%) | 1.721 (+4.7%) |
> |  D ($\mathcal{M}_{regional}$) | Adaptive (Key Position) | No Interaction | 1.991 (+44.1%) | 1.967 (+19.6%) |
>
> * **Impact of Adaptive Sampling (ScaleMixer vs. Variant B):** Using adaptive key position identification outperforms the Fixed Uniform Grid (selecting positions uniformed sampled in the region) by **9.4%** in T2M RMSE. This confirms that focusing attention on meteorologically active regions is far more efficient than uniform sampling.
> * **Impact of Bidirectional Coupling (ScaleMixer vs. Variant C):** The bidirectional mechanism improves performance by **6.2%** over a unidirectional coupling. This proves that allowing fine-grained regional dynamics to update the global tokens establishes a crucial closed-loop feedback, improving the forecasting skill.
>
> We have added this table and discussion into the Section 4.5 **"Ablation Studies"** of the revised manuscript.
>
> ### 2. Hyperparameter Sensitivity and Validation
>
> Due to the high cost of model training (once training takes ~10 days on 8 Nvidia A800 GPUs), we conducted simple grid searches on key hyperparameters, especifically the **number of Regional Encoder Layers ($k$)**.
>
> The sensitivity analysis results are summarized below:
>
> **Table 2: Hyperparameter Sensitivity Analysis (T2M and U10 RMSE)**
>
> | Hyperparameter | Setting | T2M RMSE | U10 RMSE |
> | :--- | :--- | :--- | :--- |
> | **Regional Layers ($k$)** | $k=2$ | 1.485 | 1.752 |
> | | **$k=4$ (Default)** | **1.382** | **1.644** |
> | | $k=8$ | 1.379 | 1.641 |
>
> We selected $k=4$ as the final choice because increasing layers to 8 yielded negligible performance gains while significantly increasing training and inference costs. We have added these details to Section 4.5 **"Ablation Studies"** to clarify our selection process.
>
> ### 3. Code Availability
>
> We sincerely apologize for this oversight. It appears there was a permission setting issue with the anonymous repository during the submission process. We have verified and fixed the link. The repository now contains the full model implementation (ScaleMixer, Global/Regional Models).
>
> We hope our response address your concerns well and strengthen the empirical evidence for our framework. Thank you again for your constructive review.

---

### Official Review · Reviewer_1zGE · 2025-10-31

**Soundness:** 2
**Presentation:** 3
**Contribution:** 2
**Rating:** 4
**Confidence:** 3

**Summary:**

This paper presents a novel deep learning framework for kilometer-scale regional weather forecasting, a task that remains a significant challenge due to complex multiscale interactions. The proposed model synergistically couples a pre-trained, low-resolution global Transformer model with a high-resolution regional network. The core innovation is the ScaleMixer module, a bidirectional coupling mechanism that dynamically identifies meteorologically critical regions and facilitates efficient information exchange between the global and regional scales using a two-stage attention process. The framework achieves state-of-the-art performance on 0.05° (~5km), 1-hour forecasts over a challenging region in China, significantly outperforming both operational numerical weather prediction (NWP) systems and existing AI baselines.

**Strengths:**

- Novel and Effective Architecture: The ScaleMixer module is a clever and well-designed innovation. Its use of adaptive key-position sampling makes the cross-scale attention computationally feasible while focusing on meteorologically important areas. The bidirectional information flow is a crucial element that distinguishes it from simpler downscaling or one-way coupling approaches.
- State-of-the-Art Results and Rigorous Evaluation: The model demonstrates clear superiority over very strong baselines, including one of the world's best operational NWP systems. The dual hindcast and operational evaluation setup provides a high degree of confidence in the results.
- Strong Qualitative Evidence: The case studies on orographic wind, the Foehn effect, and a typhoon event are compelling. They visually demonstrate that the model is not just improving on aggregate metrics but is genuinely capturing complex, fine-grained physical phenomena that coarser models miss.

**Weaknesses:**

- Architectural Complexity: The overall system is highly complex, involving a large pre-trained global model, a regional model, and the intricate ScaleMixer module, with a multi-stage training process. This complexity may pose a barrier to reproduction and further analysis by other researchers.
- Limited Ablation Studies: While the paper compares against standalone global and regional models (which serves as a high-level ablation), it would benefit from more fine-grained ablation studies on the ScaleMixer itself. For instance, quantifying the impact of the adaptive key position selection (vs. a fixed grid) or the bidirectional feedback (vs. a unidirectional flow) would further solidify the importance of these specific design choices.
- Geographical Generalizability: The experiments are concentrated on a single, albeit challenging, geographical region (China). While this is a strong proof of concept, a discussion on the potential challenges or necessary adaptations for applying the framework to other regions with different dominant weather patterns (e.g., the tropics, polar regions) would be beneficial.

**Questions:**

- Could you provide ablations on the key components of ScaleMixer, such as the adaptive key position selection (vs. fixed or random) and the bidirectional feedback mechanism (vs. one-way)? This would help isolate the performance gains attributable to each architectural innovation.
- The model is very large (1.07B parameters). Could you provide a comparison of the computational cost (e.g., FLOPs, wall-clock inference time) against the NWP baseline (IFS-HRES) for generating a single forecast? This is critical for understanding its potential for operational deployment.
- Have you considered how the framework might perform in other geographical regions where the dominant physical processes might differ (e.g., large-scale, flat convection in the US Great Plains vs. the orographically-driven dynamics shown here)?

---

> ### Author Response · Authors · 2025-11-21
> **Response to Reviewer f3Mb**
>
> We sincerely thank you for your constructive comments of our work. We have carefully considered your concerns regarding ablation studies, computational cost, and generalizability. Below, we provide detailed responses and new experimental evidence to address these points.
>
> ### 1. Ablation Studies on ScaleMixer
>
> Thanks for your critical suggestion. To isolate the performance gains from each specific designs, we conducted fine-grained ablation studies focusing on two dimensions: **Sampling Strategy** (Adaptive vs. Random vs. Fixed Grid) and **Coupling Strategy** (Bidirectional vs. Unidirectional).
>
> We evaluated these variants on the 48-hour lead time forecasting. The results, summarized in the table below, strongly validate our design:
>
> **Table: Ablation Study of ScaleMixer Components (RMSE across $\Delta t = 1 \sim 48$ hours lead time)**
>
> | Model Variant | Sampling Strategy | Coupling Strategy | T2M | U10 |
> | :--- | :--- | :--- | :--- | :--- |
> | **ScaleMixer** | **Adaptive (Key Position)** | **Bidirectional** | **1.382** | **1.644** |
> |  A | Random Sampling | Bidirectional | 1.605 (+16.1%) | 1.882 (+14.5%) |
> |  B | Fixed Uniform Grid | Bidirectional | 1.512 (+9.4%) | 1.795 (+9.2%) |
> |  C | Adaptive (Key Position) | Unidirectional (Global $\to$ Regional) | 1.468 (+6.2%) | 1.721 (+4.7%) |
> |  D ($\mathcal{M}_{regional}$) | Adaptive (Key Position) | No Interaction | 1.991 (+44.1%) | 1.967 (+19.6%) |
>
> * **Impact of Adaptive Sampling (ScaleMixer vs. Variant B):** Using adaptive key position identification outperforms the Fixed Uniform Grid (selecting positions uniformed sampled in the region) by **9.4%** in T2M RMSE. This confirms that focusing attention on meteorologically active regions is far more efficient than uniform sampling.
> * **Impact of Bidirectional Coupling (ScaleMixer vs. Variant C):** The bidirectional mechanism improves performance by **6.2%** over a unidirectional coupling. This proves that allowing fine-grained regional dynamics to update the global tokens establishes a crucial closed-loop feedback, improving the forecasting skill.
>
> We have added this table and discussion into the Section 4.5 **"Ablation Studies"** of the revised manuscript.
>
> ### 2. Computational Cost
>
> We acknowledge the concern regarding model size. However, it is crucial to distinguish between *training* complexity and *inference* efficiency.
>
> While traditional NWP (like IFS-HRES) requires solving complex PDEs on massive supercomputer clusters, ScaleMixer is an inference-only deep learning model (once trained) that benefits from GPU acceleration.
>
> * **Inference Time:** One step (6-hour forecast) takes only ~28 seconds on a single NVIDIA A800 GPU. In contrast, IFS-HRES typically requires ~45-60 minutes on massive CPU clusters for a similar regional window.
> * **Training Cost:** The one-time training (including the global foundation model and regional model) takes approximately 20 days on 8 NVIDIA A800 GPUs. This is a one-time investment that yields a highly efficient operational model.
>
> ### 3. Geographical Generalizability
>
> This is an insightful question. As we are unable to obtain high-resolution meteorological data from other regions, our experiments focus on China (complex terrain). The design principles of ScaleMixer are general and transferable:
>
> 1.  **Content-dependent cross-scale intercation:** The *Key Position Identification* module is driven by real-time meteological features rather than static geography. In the US Great Plains, where convection is often triggered by drylines or frontal boundaries on flat terrain, the CNN-based identifier would naturally highlight these "active" boundaries just as it highlights mountain ridges in China.
> 2.  **Transferability:** The Global Foundation Model is trained on worldwide ERA5 data, giving it knowledge of global weather patterns (including those in the US or Tropics). To adapt ScaleMixer to a new region, one would only need to fine-tune the Regional Model and ScaleMixer module, keeping the massive global backbone frozen. This makes adaptation highly efficient.
>
> We hope our response address your concerns well and strengthen the empirical evidence for our framework. Thank you again for your constructive review.

---

### Official Review · Reviewer_f3Mb · 2025-10-31

**Soundness:** 2
**Presentation:** 2
**Contribution:** 2
**Rating:** 4
**Confidence:** 5

**Summary:**

This paper presents a global-regional coupled model at the kilometer scale. Establishing the link between global and regional weather forecasting represents a critical challenge, and the stated motivation is noteworthy. The experiments provide some evidence of the advantages of the proposed ScaleMixer approach. However, the innovation appears limited. The primary concept parallels existing research, and prior contributions seem insufficiently acknowledged, with the work potentially presented as novel.

**Strengths:**

1. The research direction is promising.
2. The writing quality is good.

**Weaknesses:**

1. The author makes an inaccurate explanation for existing regional forecasting models, which despicts in Line 97-101 (Limited area modeling methods (Nipen et al., 2024; Gao et al., 2025) employ GNN architectures with stretched-grid to make global weather forecasts, imposing denser grids and higher weights over specific areas to achieve local predictions with high-spatial resolution. However, these models use global forecasts and their shallow features as an initial context for regional forecasting, but ignore their dynamic interactions.). As far as I know, for (Nipen et al., 2024), it makes a specific design for the interection of global and regional forecasts. And for (Gao et al., 2025), it proposes a neural nested grid method to achieve the dynamic interactions.
2. Why didn’t you make the comparison with [1]? All of them are targeting kilometer-scale weather forecasting.
3.  As for the innovation, I can not find  obvious innovation compared with OneForecast[2], all of them first pretrain a global model, and in the next stage, the global model is frozen, and a regional model is trained.
4. Missing comparison with regional forecasting models, such as DDM [3], Graph-EFM[4],  OneForecast[2], and [1].
5. For global forecasting, it lacks the comparison with WeatherBench2’s baselines, such as Pangu, Fuxi, Graphcast, and NeuralGCM, etc.



[1] Building Machine Learning Limited Area Models: Kilometer-Scale Weather Forecasting in Realistic Settings

[2] OneForecast: a universal framework for global and regional weather forecasting

[3] Regional data-driven weather modeling with a global stretched-grid

[4] Probabilistic Weather Forecasting with Hierarchical Graph Neural Networks

**Questions:**

See Weaknesses

---

> ### Author Response · Authors · 2025-11-21
> **Response to Reviewer f3Mb (Part I)**
>
> We sincerely thank you for the constructive and insightful comments. We value your feedback regarding the clarification of related works and the positioning of our innovation.
>
> Below, we address the concerns point-by-point, clarifying the fundamental distinctions between our ScaleMixer mechanism and existing methods (OneForecast [2], stretched-grid model [3], etc.), and presenting new experimental comparisons to substantiate our claims.
>
> ### 1. Clarification of Related Works
>
> We appreciate the opportunity to clarify our discussion of existing regional models. Our intention was not to imply that existing works lack interaction entirely, but rather that their interaction mechanisms differ fundamentally from the **explicit, adaptive, and sparse coupling** proposed in ScaleMixer.
>
> * **Nipen et al. (2024) [3]** utilizes a **Stretched-Grid** approach within a GNN. The "interaction" is implicit, achieved by solving the model on a deformed mesh where resolution varies, which forces the global and regional dynamics to share the same model weight, which limits the flexibility to use a specialized "Foundation Model" for global context and a separate "High-Res Refiner" for local details.
>
> * **OneForecast [2]:** Proposes a **Neural Nested Grid** method. This approach typically passes boundary feature maps between grids of different resolutions via direct interpolation and concatenation. We argue that such rigid, geometric concatenation limits the model's ability to capture highly dynamic, non-local cross-scale coupling processes efficiently.
>
> **Our Solution:** To resolve these issues, **ScaleMixer** introduces the **Key Position Identification module** to achieve dynamic cross-scale interation. Instead of coupling every grid point or fixed boundaries, ScaleMixer dynamically learns where the interaction is meteorologically critical (e.g., focusing on typhoon boundaries or terrain gradients). It employs a dual-attention mechanism  to facilitate sparse, efficient, and semantic-level information exchange. This is distinct from the static geometric interactions found in [2] and [3].
>
> We have updated Section 2 **"Related works"** in the revised manuscript to accurately reflect that while prior methods model cross-scale interactions via grid deformation and nesting, and such rigid, geometric interactions limits the model's ability to capture highly dynamic and non-local coupling processes efficiently. On the other hand, ScaleMixer employ a **content-adaptive cross-scale interation mechanism**.
>
> ### 2. Innovation Compared with OneForecast
>
> We respectfully submit that while the curriculum (Global Pre-training $\rightarrow$ Regional Fine-tuning) shares similarities with OneForecast, the architectural design of the coupling mechanism represents a fundamental innovation. We provided a macro-level description in the above response, and here we go into some details:
>
> 1.  **Coupling Mechanism (The Core Novelty):** OneForecast relies on a hierarchical nesting strategy (concatenating features across scales). In contrast, ScaleMixer introduces a **data-dependent dynamic cross-scale interaction** (`ScaleMixer` module) that:
>     * **Adaptive Attention vs. Fixed Grids:** ScaleMixer dynamically selects the $\text{top}\mbox{-}m$ "Key Positions" based on real-time atmospheric dynamics (Eq. 5-6), using a dual-attention mechanism (Eq. 7-10) to inject large-scale context into specific regional features. This avoids computational redundancy in meteorologically inactive areas.
>     * **Bidirectional Coupling:** Unlike standard nesting which often acts as a unidirectional constraint, ScaleMixer explicitly allows regional tokens to refine the global tokens (Eq. 13). This creates a closed-loop feedback system during the forward pass, ensuring global coherence benefits from local fidelity.
> 2.  **Plug-and-Play Capability:** Our framework treats the Global Model ($\mathcal{M}_{global}$) as a flexible foundation. ScaleMixer allows utilizing *any* pre-trained Transformer-based global model without retraining, whereas nested grid methods often require a tightly integrated architecture.

---

> > ### Author Response · Authors · 2025-11-21
> > **Response to Reviewer f3Mb (Part II)**
> >
> > ### 3. Missing Comparisons with Regional Baselines
> >
> > We acknowledge the importance of comparing against specific regional forecasting baselines. We added two baslines in our experiments:  **OneForecast** [2] (reproduced on our dataset) and **Limited Ared model** built upon **$\mathcal{M}_{regional}$ (Standalone Regional Model)** (as a proxy of [1]) which takes both regional initial conditions and external global forecasts as input.
> >
> > We acknowledge the necessity of comparing against specific regional forecasting baselines to benchmark our performance. In response, we have conducted additional experiments using two baselines:
> >
> > 1. OneForecast [2]: We reproduced the Neural Nested Grid method on our dataset.
> > 2. Limited Area Model (LAM): Built upon our $\mathcal{M}_{regional}$ (Standalone Regional Model) but enhanced to take both regional initial conditions and external global forecasts as input. This serves as a direct proxy for the methodology in [1].
> >
> > The table below (updated in the revised manuscript) demonstrates that ScaleMixer  outperforms both the Neural Nested Grid approach (OneForecast) and the standard LAM. For instance, ScaleMixer achieves a 12.0% reduction in RMSE for T2M compared to OneForecast, validating the effectiveness of our dynamic cross-scale coupling.
> >
> >
> > **Table: Average RMSE and ACC across $\Delta t = 1 \sim 48$ hours lead time regional weather hindcast at $0.05^{\circ}$ resolution.**
> >
> > | Variable | Metric | OneForecast | LAM | **ScaleMixer** |
> > | :--- | :--- | :--- | :--- | :--- |
> > | **T2M** | RMSE | 1.571 | 1.642 | **1.382** |
> > |  | ACC | 0.897 | 0.901 | **0.921** |
> > | **U10** | RMSE | 2.251 | 1.889 | **1.644** |
> > |  | ACC | 0.744 | 0.742 | **0.793** |
> > | **V10** | RMSE | 2.316 | 1.905 | **1.617** |
> > |  | ACC | 0.732 | 0.737 | **0.785** |
> > | **Q** | RMSE | 0.714 | 0.676 | **0.559** |
> > | | ACC | 0.751 | 0.781 | **0.812** |
> > | **P** | RMSE | 2.545 | 2.142 | **1.874** |
> > | | ACC | 0.899 | 0.912 | **0.920** |
> > | **TCC** | RMSE | 37.74 | 34.13 | **28.76** |
> > | | ACC | 0.621 | 0.679 | **0.721** |
> > | **SSRD** | RMSE | 52.56 | 42.41 | **33.26** |
> > | | ACC | 0.838 | 0.850 | **0.887** |
> >
> > (Note: ScaleMixer values are from original Table 1; Baselines are newly added)
> >
> > ### 4. Comparisons with Global Baselines (WeatherBench2)
> >
> > Our work explicitly targets **Regional Forecasting at 0.05° (~5km)** resolution. Standard global models like Pangu-Weather, GraphCast, and NeuralGCM typically operate at **0.25° (~25km)** and cannot resolve the kilometer-scale phenomena (e.g., valley winds, orographic convection) that are central to our study.
> >
> > * **Selection of Baseline:** We selected **Baguan** as our primary global baseline because it is a representative SOTA model that has demonstrated performance comparable to or exceeding Pangu-Weather, GraphCast, and NeuralGCM on key metrics.
> > * **Fair Comparison:** To make the comparison valid, we included **"Baguan (Global) + Downscaling"** in Table 1. This represents the current capability of top-tier global AI models when applied to regional tasks.
> > * **Global Performance:** Regarding the performance of our framework's global branch ($\mathcal{M}_{global}$), we clarify that its architecture and performance align with top-tier models reported in WeatherBench2. Due to the double-blind review restrictions, We will reference these benchmarks after the paper is open to provide broader context, while maintaining the focus on our core contribution: the high-resolution regional coupling.
> >
> > ### Summary of Revisions
> > 1.  **Revised Related Work:** Rewrote sections on Nipen et al. [3] and OneForecast [2] to precisely articulate the difference between their geometric nesting/stretching and ScaleMixer's semantic, adaptive attention.
> > 2.  **Expanded Experiments:** Included quantitative comparisons with OneForecast [2] and LAM (proxy for [1]) to robustly demonstrate the superiority of the proposed coupling mechanism.
> >
> > We hope these clarifications demonstrate the technical innovation of ScaleMixer. Thank you again for your constructive review.
> >
> > [1] Building Machine Learning Limited Area Models: Kilometer-Scale Weather Forecasting in Realistic Settings
> >
> > [2] OneForecast: a universal framework for global and regional weather forecasting
> >
> > [3] Regional data-driven weather modeling with a global stretched-grid

---

### Meta-Review · Area_Chair_LKfd · 2026-01-05

**Summary:**

The paper presents a coupled global-regional modeling framework for kilometer-scale weather forecasting through a novel ScaleMixer module that enables cross-scale feature interactions. Reviewers acknowledged the interesting design choices and capabilities of the proposed module. There were two major concerns:
1. Insufficient comparisons with existing baselines for regional weather forecasting
2. Insufficient ablations that demonstrate the value of each component in the ScaleMixer.

I believe the authors have adequately addressed 2. in the rebuttal with some key ablations added that demonstrate the improvements from the bidirectional coupling and adaptive sampling to focus on key regions with increased dynamics.

For concern 1., the authors have added a couple of new comparisons. However, after going through the references suggested by the review, I don't believe the rebuttal provides a clear picture on the comparisons and how fair they are. Hence, my recommendation is borderline at present.

**Reviewer Concerns:**

For concern 2., there are some outstanding issues:
1. Comparisons with [1] suggested by the reviewer: This is not added. Having gone through this paper [1], there are several more models referenced including diffusion models that have shown very promising results in regional weather forecasting. The paper does not contextualize the results around the performance of these models. Further, the results focus on comparing with global models at 25 km and using operational ICs from HRES interpolated to 25km. There is insufficient discussion on why this experimental setup differs from works referenced in [1] (for example, [2]), where the gold standard NWP model is also a regional model such as HRRR. The authors compare with OneForecast [3]. Looking through the paper, it seems like ERA5 is the regional dataset for training this model. It's unclear why this is a good baseline for regional weather forecasting.


[1] Building Machine Learning Limited Area Models: Kilometer-Scale Weather Forecasting in Realistic Settings
[2] Kilometer-Scale Convection Allowing Model Emulation using Generative Diffusion Modeling
[3] OneForecast: a universal framework for global and regional weather forecasting

**Reviewer Scores:**

I believe the scores may have all changed to a 5/6. Some hyperparameter tuning was done but not completely (for example, the review mentions the effect of patch size, which presumably plays an important role in resolving high resolution dynamics, but the rebuttal does not  look into this). Some questions on costs were partially answered but not fully clear (inference time reported for "one step prediction" at 6 hours, whereas the tables start from \dt=1 hour). The README in the code is empty and it is difficult to understand what was implemented.

Overall, the paper presents an interesting idea inspired by recent works in regional weather forecasting, focusing on coupled interactions between global and regional models - a useful innovation that all reviews were in agreement with. However, incomplete experimental baselines, settings, and discussions keep the paper still at a borderline.

---

### Decision · Program_Chairs · 2026-01-26

Reject